# `LeMaRns`: A Length-based Multi-species analysis by numerical simulation in R

**Michael A. Spence**●*, **Hayley J. Bannister, Johnathan E. Ball, Paul J. Dolder**●,
**Christopher A. Griffiths, Robert B. Thorpe**

Centre for Environment, Fisheries and Aquaculture Science, Lowestoft, Suffolk, United Kingdom

* michael.spence@cefas.co.uk

## Abstract

Fish stocks interact through predation and competition for resources, yet stocks are typically managed independently on a stock-by-stock basis. The need to take account of multi-species interactions is widely acknowledged. However, examples of the application of multi-species models to support management decisions are limited as they are often seen as too complex and lacking transparency. Thus there is a need for simple and transparent methods to address stock interactions in a way that supports managers. Here we introduce `LeMaRns`, a new R-package of a general length-structured fish community model, LeMans, that characterises fishing using fleets that can have different gears and species catch preferences. We describe the model, package implementation, and give three examples of use: determination of multi-species reference points; modelling of mixed-fishery interactions; and examination of the response of community indicators to dynamical changes in fleet effort within a mixed-fishery. `LeMaRns` offers a diverse array of options for parameterisation. This, along with the speed, comprehensive documentation, and open source nature of the package makes LeMans newly accessible, transparent, and easy to use, which we hope will lead to increased uptake by the fisheries management community.

**Data Availability Statement:** All relevant data are within the paper and its Supporting Information files. he R-package is available on CRAN (https://cran.r-project.org/web/packages/LeMaRns/index.html).

## Introduction

In fisheries management, fish stocks are typically managed on a stock-by-stock basis, with most assessment models having a single-species focus that does not explicitly model interactions between stocks. Single-species models are useful for assessing the current state of a stock and making short-term forecasts as you only need information regarding the stock in question. However, these models assume that inter-stock interactions (predation and competition for resources) are either fixed or vary only in a simple way with time, and so on longer timescales their use becomes increasingly problematic. Multi-species models that explicitly represent some or all of these interactions are generally more suited to making predictions as the time-scale of interest increases [1]. As a result, a large number of multi-species models have been developed. The models take various approaches: statistical models, such as the Stochastic Multi-Species model (SMS) [2], are similar to age-structured single-species assessment models;

**Funding:** MAS, HJB, JEB, PJD, CAG and RBT were all funded by the European Union's Horizon 2020 research and innovation programme under Grant Agreement No 773713 (PANDORA). No the funders had no role in study design, data collection and analysis, decision to publish, or preparation of the manuscript.

biomass dynamic models, such as surplus-production models [3], describe the dynamics of bulk biomass; while mechanistic models, such as Ecopath [4], attempt to describe the processes that lead to the emergent system based on ecological theory. For a discussion regarding the relative merits of single and multi-species approaches see [1].

Size-based multi-species models are a class of mechanistic models [5] that, in contrast to single-species models, explicitly take account of foodweb interactions. In marine ecosystems size is often a key trait [6]. By representing many processes, including fishing, natural mortality, and predation as a function of length, it is possible to reproduce many aspects of the community dynamics (such as the tendency of diet to change with increasing predator size; [7]) with a relatively small number of parameters and a modest requirement for data in model set up. This makes the framework particularly suitable for use in data-limited fisheries. However, size-based multi-species models lack uptake by both managers and fisheries scientists due to the perceived complexity and lack of transparency of the models [8].

Here we introduce the new `LeMaRns` R-package, within which we implement a general length-structured fish community model, LeMans (Length-based Multi-species analysis by numerical simulation) [9, 10], that can be adapted to any marine ecosystem with only a modest amount of data. The speed, increased functionality, comprehensive documentation, diverse array of parameterisation, and open source nature of the package makes LeMans newly accessible to model developers and general users alike. We hope this will lead to increased user uptake and novel model development by the fisheries community.

In this paper we give a brief overview of the `LeMaRns` model, we describe how the package works and we provide three applications of `LeMaRns`. The code for these applications can be found in S1 File. We conclude with a discussion regarding future work.

## Model overview

The LeMans model framework was originally developed by [9] to represent the Georges Bank fish community, but was subsequently adapted for use in the North Sea by [10–14].

LeMans is well suited for use whenever there is a need for multi-species or mixed-fisheries analysis but where there is insufficient data to support the use of more complicated models, such Atlantis [15]. The model has been used to assess the impact of mixed-fisheries [12, 13], evaluate the effect of harvest control rules [14], and as part of a multi-model ensemble along with other multi-species models [16]. LeMans models fish length because a) it is generally easier to measure than weight in the field [17], and b) fisheries selectivity is normally characterised in terms of length [18] and is thus more straightforward to relate to the parameterisation of mixed-fisheries.

The model has no spatial dependency and describes the dynamics of multiple species in $n_l$ discrete length classes through time. A year in the model is subdivided into a number of equal time steps of length $\delta t$. Spatial information can be included in the model implicitly via the predator-prey interaction matrix (e.g. [19]).

Let $N_{j,i,t-1}$ be the number of individuals of the $i$th species in the $j$th length class after $t-1$ time steps. During each time step three processes occur: recruitment, mortality, and growth. The number of individuals after the recruitment phase of the $t$th time step is

$$N'_{j,i,t} = \begin{cases} N_{j,i,t-1} + R_{i,t} & \text{if } j = 1 \\ N_{j,i,t-1} & \text{otherwise,} \end{cases}$$

where $R_{i,t}$ is the number of recruits of the $i$th species at time $t$. $R_{i,t}$ depends on the spawning stock biomass of the $i$th species as well as the time step of the model. In [9] and [10]

recruitment occurred in the first time step of a new year, using the a Ricker recruitment curve [20] and a hockey-stick recruitment curve [21] respectively (see S2 File pages 5-6 for more details).

The number of individuals after the mortality phase of the time step is

$$N_{j,i,t}^{''} = N_{j,i,t}^{'} \ \exp(-(M1_{j,i} + M2_{j,i,t} + F_{j,i,t})\delta t).$$

where $M1_{j,i}$ is the background mortality, $M2_{j,i,t}$ is the predation mortality, and $F_{j,i,t}$ is the fishing mortality. The background mortality, $M1_{j,i}$, is size and species dependent (see S2 File page 7 for more details). Predation mortality is size- and species-dependent. The size preference of a predator is described using a preference function based upon a log-normal distribution, whilst species preference is described using a predator-prey interaction matrix indicating who eats whom [10, 11] (see S2 File pages 4, 7-8 and 13 for more details).

Fishing mortality is constructed from the joint effect of a number of fishing gears exploiting different species with different size and targeting preferences. More specifically, the fishing mortality of the $i$th species in the $j$th length class at time step $t$ is

$$F_{j,i,t} = \sum_{k=1}^{H} e_{k,t}q_{j,i,k}, \qquad (1)$$

where $e_{k,t}$ is the effort of the $k$th gear, for $k = 1\ldots H$ (the total number of gears in the fishery) and $q_{j,i,k}$ is the catchability. The units for $F$ are $yr^{-1}$ and the units of $q_{j,i,k}$ are $F$ per unit effort.

The number of individuals after the growth phase, and the end of the time step, is

$$N_{j,i,t} = \begin{cases} N_{j,i,t}^{''}(1 - \phi_{j,i}) & \text{if } j = 1 \\ N_{j,i,t}^{''}(1 - \phi_{j,i}) + N_{j-1,i,t}^{''}\phi_{j-1,i} & \text{otherwise,} \end{cases}$$

where $\phi_{j,i}$ is the proportion of individuals of species $i$ that leave length class $j$ due to growth over the time step according to the von-Bertalanffy growth equation [22] (see S2 File pages 3-4 for more details). Further details of the model can be found in S2 File (pages 2-14) and in [9].

## Using `LeMaRns`

LeMaRns is available on CRAN (https://cran.r-project.org/web/packages/LeMaRns/index.html) and GitHub (https://github.com/CefasRepRes/LeMaRns).

## Data requirements

**Biological data.**  The minimum amount of information required to set up a model using `LeMaRns` includes: species-specific maximum length (`Linf`), length at 50% maturity (`Lmat`), length-weight conversion parameters (e.g. `W_a` and `W_b`), the growth parameter from the specialised von Bertalanffy growth function (`k`) [23], and the recruitment parameters (`recruit_params`; see S2 File, page 5 for further details). The length-weight and life history parameters are often available from survey data, online databases (e.g. Fishbase [24]), or through 'life history invariants' [25, 26]. The recruitment parameters are typically harder to determine and can be thought of as 'tuning' parameters [6]. The fitting of these parameters is done outside of the `LeMaRns`. An illustrative example, based on the methods in [27], can be found in S3 File.

In addition to the required parameters described above, users may specify species-specific values for $M1$, background mortality (i.e. mortality not from fishing or predation), and the rate of change from immaturity to maturity, although default values are given for these. Users may also input a predator-prey interaction matrix, `tau`, which describes the diet information

and spatial overlap of predators and prey. `tau` defaults to one for all predator-prey combinations, although we recommend that this is replaced with an ecosystem-specific matrix based on available diet information, spatial overlap, and/or expert judgement.

In `LeMaRns` there are five built-in recruitment functions: `hockey-stick` [21] (the default option), `Ricker` [20], `Beverton-Holt` [23], `linear`, or `constant`, as well as three background mortality functions: `std_RNM` (the default option), `constant`, and `linear` (see S2 File, pages 5 and 7).

The predator-prey mass ratio, the width of the predator-prey size preference, and the theoretical growth efficiency of a fish of length zero are all species-independent parameters in the current version of `LeMaRns`.

**Fishing.** `LeMaRns` allows mixed-fisheries analyses to be conducted through the definition of fishing gears. In the LeMans model, the fishing mortality is calculated using Eq 1. This means that the catchability, $q_{j,i,k}$, for the $i$th species in the $j$th length class with the $k$th gear must be defined for all species, length classes, and gears. The catchability is fixed in time but effort, $e_{k,t}$, can be dynamic. In `LeMaRns` there are three built-in functions that can be used to create catchability curves: `logistic`, `log_gaussian`, and `knife-edge` (see S2 File, page 8). In addition, there is an option that allows users to input their own catchabilities.

**Test dataset.** In `LeMaRns`, we provide a dataset, `NS_par`, for 21 species in the North Sea based on [10]. The dataset contains `Linf`, `Lmat`, `W_a`, `W_b`, `k`, and the recruitment parameters (`a` and `b`) for each species. We also include `NS_other` to represent other food. The recruitment parameters, `a` and `b`, and `NS_other` were calibrated to the North Sea (see S3 File for details). The predator-prey matrix (`NS_tau`) contains information regarding the diet of the 21 species and is based on [10]. In addition, we provide information regarding a number of fishing fleets (`NS_mixed_fish`), with catchability parameters (`NS_eta` and `NS_L50`) that are based on [12].

## Setting up the model

In `LeMaRns` a model can be set up using the `LeMansParam()` function. This function returns an object of class `LeMans_param`, which contains all of the information required to run the LeMans model. Below is an example of how to use the provided data to set up the model:

```
NS_params <- LeMansParam(NS_par, tau = NS_tau,
                         eta = NS_eta, L50 = NS_L50,
                         other = NS_other)
```

The `LeMansParam()` function takes the parameters described in the previous section, as well as optional inputs including: $n_l$, the number of length classes (`nsc`, the default is 32); the boundaries of the length classes (`bounds`, the default depends on `max(Linf)`), and $\delta t$, the time step of the model in years (`phi_min`, the default is 0.1).

All default values, with the exception of `tau`, are the same as those used in [10].

## Running the model

In `LeMaRns` a model can be run using the `run_LeMans()` function:

```
run_LeMans(NS_params)
```

By default, `run_LeMans()` uses the `get_N0()` function to initialise the population and is run for for 10 years with no fishing. However, users can specify their own initial population with the input `N0`, a feature that also allows users to extend model runs when required (see S2 File for examples).

Although gear catchability is calculated in `LeMansParam()`, fishing effort is an input to `run_LeMans()`, thus allowing effort to be dynamic. `years` is also an input to `run_LeMans()` and is used to define the number of years that the model should be run for. Below we run the model for 50 years with a constant effort of 0.25, which equates to an *F* of 0.25 on the length-class with highest selection, for each gear:

```
no_of_gears <- dim(NS_params@Qs) [3]
effort_mat <- matrix(0.25, 50, no_of_gears)
model_run <- run_LeMans(NS_params, years = 50, effort =
effort_mat)
```

## Model outputs

`run_LeMans()` returns an object of class `LeMans_output`, which contains a time series of the number of individuals in each length class for each species and time step (`N`), the weight caught in each length class for each species and time step (`Catch`), the predation mortality in each length class for each species and time step (`M2`), and the number of recruits of each species and time step (`R`).

The `LeMaRns` package also includes a number of built-in functions that enable users to explore the outputs of a model run in more detail. These functions can be used to calculate and plot community and species-specific total biomass and Spawning Stock Biomass (SSB), as

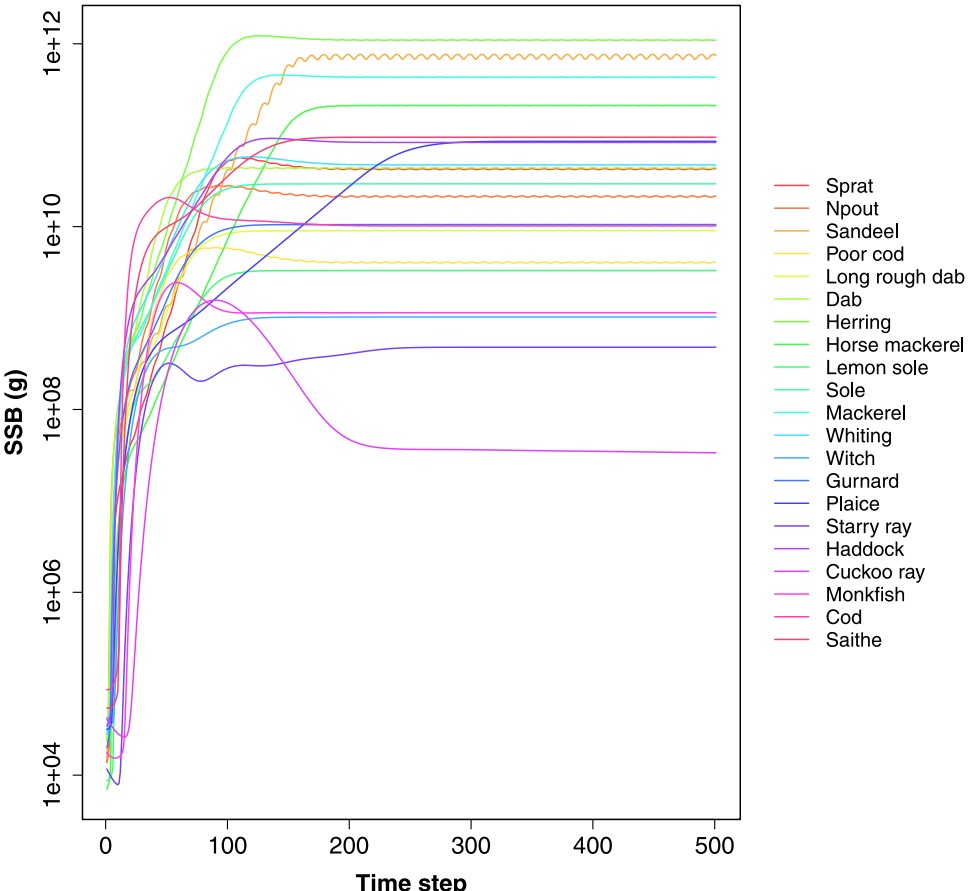

**Fig 1. The Spawning Stock Biomass (SSB) plotted using the `plot_SSB()` function.**

well as several ecosystem indicators including the Large Fish Indicator (LFI), Mean Maximum Length (MML), Typical Length (TyL), and Length Quantiles (LQ). Functions also exist to calculate Catch Per Unit Effort (CPUE) and Catch Per Gear (CPG). See S2 File, page 18 for definitions of these outputs.

An example is shown in Fig 1, which is created using `plot_SSB(NS_params, model_run)`.

## Case studies

In this section we provide three applications of the `LeMaRns` package. The first example focuses on finding long-term multi-species fishing targets; the second examines the effect of different fishing scenarios on long-term stock status in a mixed-fishery; and the third example explores the effect of dynamic fishing effort in a mixed-fishery on ecosystem indicators. In the applications we assume that each species is a single stock. The code for generating these examples can be found in S1 File.

### Nash equilibrium

Fish stocks are often managed by considering the fishing mortality that maximises the long-term yield, i.e. the Maximum Sustainable Yield (MSY) [28]. We can define $f_i(F_i, \boldsymbol{F_{-i}})$ as the $i$th stock's long-term yield, where $F_i$ is the fishing mortality of the $i$th stock and $\boldsymbol{F_{-i}}$ are the fishing

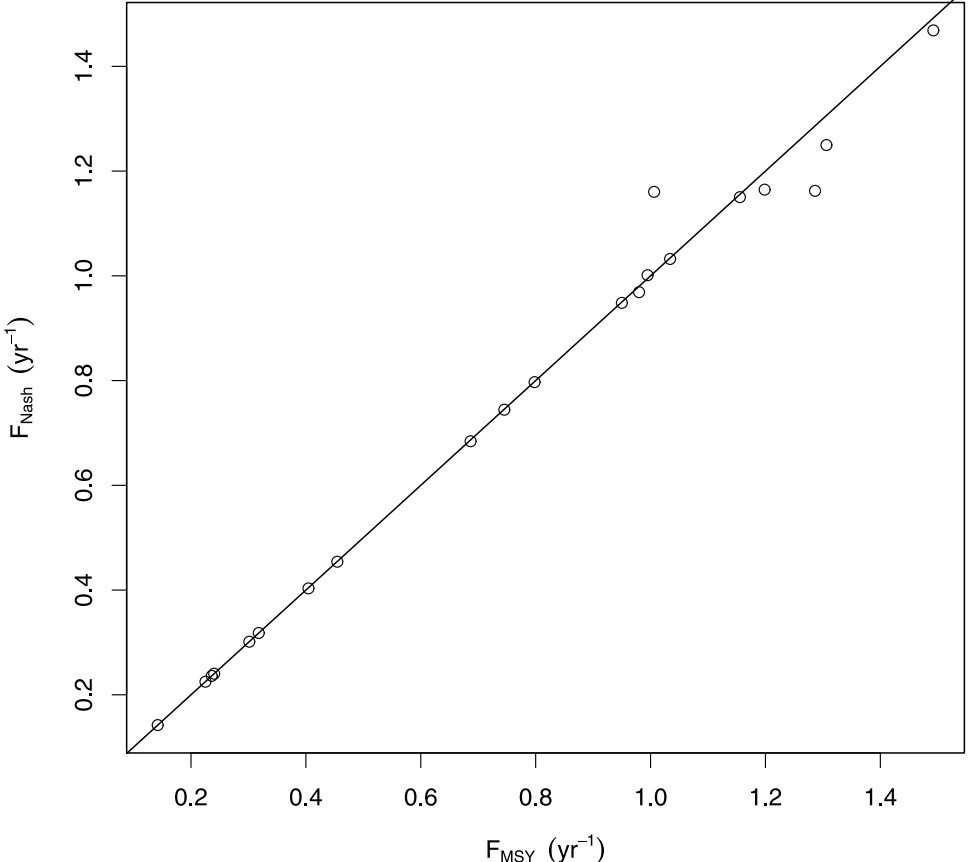

**Fig 2. $F_{MSY}$ and $F_{Nash}$ calculated using the LeMans model for the 21 species.** The solid line is the 1-1 line.

mortalities of the other stocks. Many stocks are managed on a stock-by-stock basis using single-species models. This means that

$$f_i(F_i, \boldsymbol{F}_{-i}) = f_i(F_i, \boldsymbol{F}'_{-i}),$$

$\forall \boldsymbol{F}'_{-i}$, and then

$$F_{MSY,i} = \arg \max_{F_i} f_i(F_i, \boldsymbol{F}_{-i})$$

is commonly well defined. However, stocks often interact with one another and the fishing mortality of the $j$th stock affects the catch of the $i$th stock, i.e.

$$f_i(F_i, \boldsymbol{F}_{-i}) \neq f_i(F_i \boldsymbol{F}'_{-i}).$$

We therefore need to define a multi-species MSY. One possibility is the Nash equilibrium [13], which is defined as the point at which we are unable to increase $f_i(F_i, \boldsymbol{F}_{-i})$ by changing $F_i$ only, $\forall i$. Formally, $F_{Nash,i}$ is a Nash equilibrium when

$$\forall i, F_i : f_i(F_{Nash,i}, \boldsymbol{F}_{Nash,-i}) \geq f_i(F_i, \boldsymbol{F}_{Nash,-i}).$$

Using `LeMaRns` and starting from the $F_{MSY}$ values given in [12], we can find $F_{Nash,i}$ for

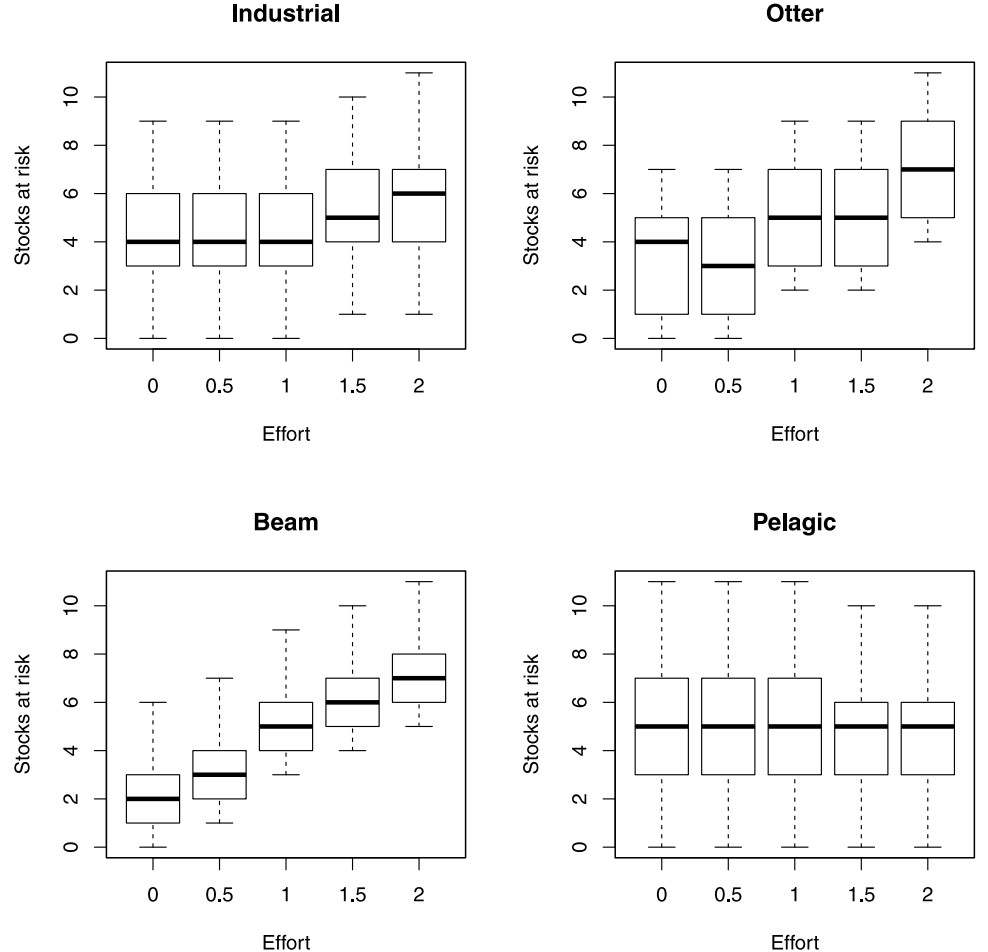

**Fig 3. The effect of varying fishing effort on the number of stocks at risk of collapse.**

$i = 1, \ldots, 21$. We can also find $F_{MSY,i}$ values to compare to our $F_{Nash,i}$; this is not trivial as we need to define fishing mortalities for all of the species and $F_{MSY,i}$ will be sensitive to these. Arbitrarily, we can set the fishing mortality for the other species to the values given in [12]. Fig 2 provides a comparison between $F_{MSY,i}$ and $F_{Nash,i}$; they appear to be similar for the species with lower $F_{MSY,i}$ and $F_{Nash,i}$, but differ more for larger values.

In this study we arbitrarily chose to hold the fishing mortality of the other stocks at the $F_{MSY}$ values given in [12]. However, if we had chosen to hold them at $F_{Nash,-i}$, then $F_{MSY,i} = F_{Nash,i}, \forall i$, as $F_{Nash,i}$ is a solution of $F_{MSY,i}$. This highlights the sensitivity of $F_{MSY,i}$ to the fishing mortality on the other stocks.

## Mixed-fishery

Here we explore the mixed-fishery example described in [12], which involves four idealised fishing fleets, i.e. a single species is caught by only one gear type. We investigated the risk of stock collapse under different fishing scenarios. A stock is deemed to have collapsed if its SSB falls below 10% of its unfished SSB [12, 29].

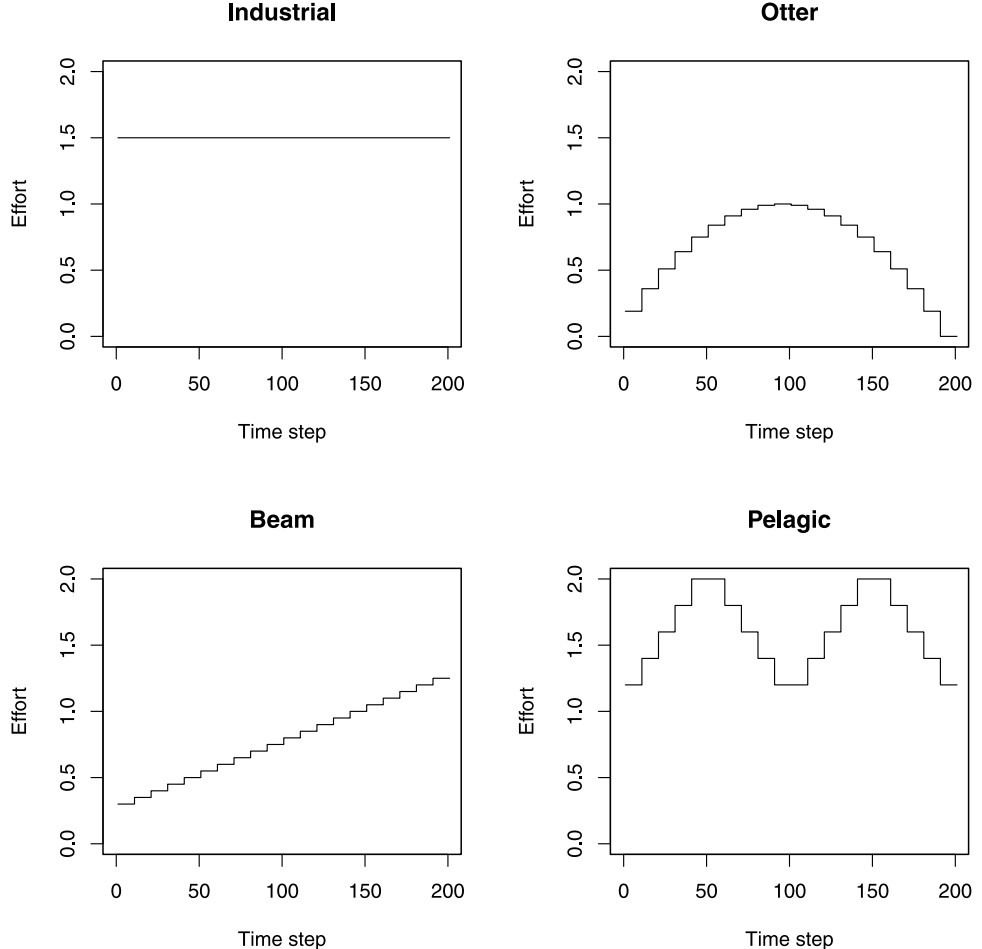

**Fig 4. The effort for the four fishing fleets.** The recreational fishing fleet is not shown but increases linearly from 0.1 in the first year to 0.15 in the final year.

The dataset `NS_mixed_fish` contains information on which fleet catches which species. In this example, the selectivity of each species follows the `logistic` curve with catchability parameters `eta` and `L50`.

In a scenario, the effort of each of the four fishing fleets, `Industrial`, `Otter`, `Beam`, and `Pelagic`, was one of five levels, `c(0,0.5,1,1.5,2)`, which was held constant for 50 years. We ran all possible combinations of these levels resulting in 625 different scenarios.

Fig 3 depicts the number of stocks at risk under varying levels of fishing effort for each fleet; the number of stocks at risk of collapse is mostly sensitive to the effort of the `Otter` and `Beam` fleets.

## Dynamic fishing

Here we add another fleet (`Recreational`) to the idealised fleets in the previous example. This fleet is set up to catch cod, haddock, herring, horse mackerel, mackerel, plaice, saithe, and whiting, with all fish exceeding the minimum landing size [30] being retained. Any fish that are discarded are assumed to have survived (following a `knife-edge` selectivity function). Having run the LeMans model for 50 years with no fishing, the model was run for a further 20 years with dynamical fishing effort (see Fig 4 and S2 File, page 39 for the time series of fishing effort).

Fig 5 depicts the MML, TyL, the LFI (40cm threshold), and the 0.1, 0.5, and 0.9 LQs in the last 20 years of the model run. The different fishing fleets seem to have a different effect on the

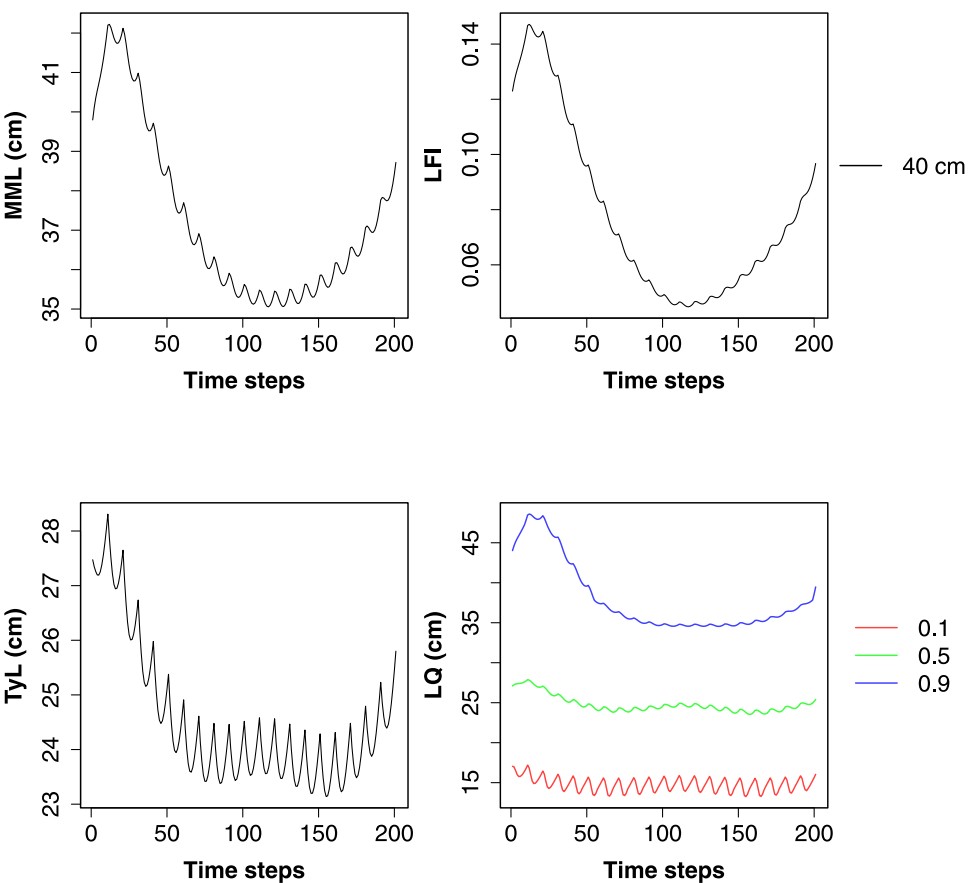

**Fig 5. The effect of fishing scenario on community indicators.** This plot was created using the `plot_indicators()` function.

dynamics of the indicators. MML and the LFI seem to correlate with the fishing effort of the `Otter` fleet, whilst TyL is additionally affected by the `Pelagic` fleet. The dynamics of the LQs suggest that larger fish are affected by the `Otter` fleet and medium sized fish by the `Pelagic` fleet. The smaller fish do not have a large reaction to fishing, but seem to have inter-annual variation due to spawning.

## Conclusions and further work

`LeMaRns` provides a convenient and user friendly way to run the LeMans model, with comprehensive documentation. The package contains the required functions to explore different fishing scenarios in a mixed-fisheries multi-species model and allows for the customisation and tailoring of inputs to model specific environments, conditions, and scenarios. This, along with the low data requirements, makes `LeMaRns` a transparent, easy to use, and broadly applicable fisheries assessment tool that encourages model development and experimentation. Further, we hope this will lead to an increased uptake of LeMans by the fisheries management community.

Several developments are planned for future releases, including food-dependent growth and stochastic recruitment. The package is currently being used to explore the effects of harvest control rules in the North Sea and to explore seasonal effects using a similar method to [31].

## Supporting information

**S1 File. R script.** The R script to run the case studies and generate the figures in the paper.
(R)

**S2 File. R package vignette.** Contains the description of the model and further explanation of the package.
(PDF)

**S3 File. An example of calibrating the model.** Contains an example of calibrating the model with robust uncertainty quantification using a Bayesian framework.
(PDF)

## Acknowledgments

This work was funded by the European Union's Horizon 2020 research and innovation programme under Grant Agreement No 773713 (PANDORA). We would also like to thank Gary Saggers, Finlay Scott, Sean M. Lucey and Athanassios C. Tsikiras for invaluable comments on an earlier version of the paper.

## Author Contributions

**Conceptualization:** Michael A. Spence, Hayley J. Bannister, Robert B. Thorpe.

**Formal analysis:** Michael A. Spence, Hayley J. Bannister, Johnathan E. Ball.

**Funding acquisition:** Robert B. Thorpe.

**Investigation:** Michael A. Spence, Hayley J. Bannister.

**Methodology:** Michael A. Spence, Hayley J. Bannister, Johnathan E. Ball, Robert B. Thorpe.

**Project administration:** Michael A. Spence, Robert B. Thorpe.

**Software:** Michael A. Spence, Hayley J. Bannister, Johnathan E. Ball, Paul J. Dolder, Christopher A. Griffiths.

**Visualization:** Michael A. Spence, Hayley J. Bannister, Johnathan E. Ball.

**Writing – original draft:** Michael A. Spence, Hayley J. Bannister.

**Writing – review & editing:** Michael A. Spence, Hayley J. Bannister, Johnathan E. Ball, Paul J. Dolder, Christopher A. Griffiths, Robert B. Thorpe.

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
