## [Decision Letter · Decision Letter 0]

27 Aug 2019

PONE-D-19-20184

LeMaRns: a Length-based Multi-species analysis by numerical simulation in R

PLOS ONE

Dear Mr Spence,

Thank you for submitting your manuscript to PLOS ONE. After careful consideration, we feel that it has merit but does not fully meet PLOS ONE’s publication criteria as it currently stands. Therefore, we invite you to submit a revised version of the manuscript that addresses the (mostly moderate) points raised during the review process.

Most of the comments raised by the two reviewers concern clarifications on the model or the manuscript and should be fully addressed.

We would appreciate receiving your revised manuscript by Oct 11 2019 11:59PM. To enhance the reproducibility of your results, we recommend that if applicable you deposit your laboratory protocols in protocols.io, where a protocol can be assigned its own identifier (DOI) such that it can be cited independently in the future. For instructions see: http://journals.plos.org/plosone/s/submission-guidelines#loc-laboratory-protocols

We look forward to receiving your revised manuscript.

Kind regards,

Athanassios C. Tsikliras

Academic Editor

PLOS ONE

Journal Requirements:

Reviewers' comments:

Reviewer's Responses to Questions

**Comments to the Author**

1. Is the manuscript technically sound, and do the data support the conclusions?

Reviewer #1: Yes

Reviewer #2: Partly

2. Has the statistical analysis been performed appropriately and rigorously? 

Reviewer #1: Yes

Reviewer #2: I Don't Know

3. Have the authors made all data underlying the findings in their manuscript fully available?

Reviewer #1: Yes

Reviewer #2: Yes

4. Is the manuscript presented in an intelligible fashion and written in standard English?

Reviewer #1: Yes

Reviewer #2: Yes

5. Review Comments to the Author

Reviewer #1: General comments:

The manuscript describes the implementation of the LeMans model as package for R, LeMaRns. It includes several examples to demonstrate how the package can be used. Most of the details on the model and its implementation are in the accompanying vignette. The paper is generally well written and the example code seems to do what it is supposed to do. I think that the manuscript is generally suitable for publication but there are some relatively small issues that I would like to see first addressed.

I appreciate that the vignette provides most of the details about the model and how to use the package. However, some more model details in the main manuscript would be good so that readers didn’t have to go digging too far to find out some of the basic model assumptions. For example, is the model single area? I assume so given the use of the predator-prey matrix for spatial overlap, i.e. a proxy for multiple areas.

Additionally, the time step of the model is not clear from reading the manuscript. For example, the equation for fishing mortality is described using time step (line 95), but the arguments for the *run_LeMans()* function includes year. Can the user determine the size of the time step? I see in the vignette that this is possible by using the *phi_min* argument but a note about this could be added to the main text.

Following this, there is some inconsistency in how the time axes are labelled. The axes for Figures 1 and 5 are in time steps (the scale implying 10 time steps per year), whereas the axis for Figure 4 is year.

Line 105, the predator-prey matrix, NS-tau, is described as containing information regarding the diet of the species in the model. It would be good to know what kind of information this is and how users might go about getting this information for their own models.

The supplementary R script works as it is supposed to and recreates the figures in the manuscript (tested using R 3.6.1). I note that the model does run very fast (certainly quicker than the initial implementation of the mizer model used to).

Specific points

----------------

* Line 6: “However, over longer timescales models that take account of multi-species interactions, such as predation and competition for resources are required for meaningful predictions.”

This is a strong statement. I don’t necessarily disagree with it but it would be good see to some additional exploration and justification of this point. For example, it is not always the case that multi-species models are better than single-species models for “meaningful” predictions. Multi-species models generally have higher data requirements, make more assumptions and are more difficult to parameterise than single-species models. This can result in an increase in uncertainty in the model predictions. Depending on the type of advice that is required for management it may be better to use a simple model, whose limitations are well understood, than a more complex model. Additionally, multi-species interactions may only become important when fishing mortality is relatively low, i.e. when fishing pressure is high it is the dominant source of mortality, in which case single-species models may be adequate. There are many challenges when using multi-species models, including LeMans, for example the estimation of an initial population abundance (as noted in the manuscript, which states that the provided data for the North Sea ecosystem has not been calibrated). This is simpler with a single-species model as they can often be used for stock assessment as well as projections.

I understand that the focus of the paper is not about the pros and cons of single- vs multi-species models but I think the introduction could benefit from a small discussion about when using a multi-species model might be appropriate and the kind of management advice it is able to provide, including the robustness of that advice.

* Line 8: “…several multi-species models have been developed”.

There are certainly more than several multi-species models. Do the authors mean several multi-species modelling approaches?

* Line 97: “…catch at length…”

Do the authors mean “catchability at length”?

* Line 108: “Note that due to the generalisations of the LeMans model in the LeMaRns R-package, the provided data is not calibrated to the North Sea ecosystem and is therefore used for demonstration purposes only.”

This is slightly concerning. Does this means that the implementation here is not the full LeMans model? What are these generalisations and what are the key differences between the LeMans model (as described and reviewed in references 8 – 13) and this implementation as LeMaRns? How do they impact the data in the package so that it is essentially uncalibrated?

It’s a shame that the data has not been calibrated. Some notes on how calibration could be performed would be good, rather than just pointers to references, i.e. what data is required to perform tuning?

* Line 117: Is it necessary to include the *other* argument? It isn’t described in the main text and the default value (according to the man page for LeMansParam()) is 1e12 anyway.

* Figure 1: There should be units on the y-axis for SSB.

* Figure 2: Why are the axes inverted (i.e. go from high to low)? Also, there should be units on these axes.

* Line 186: “Using a factorial design…”

It is not immediately clear what this means. I can figure it out from the attached R script but this could be expanded in the text to add clarity.

* There is a possible mistake in the package vignette. On page 13 of the vignette it says “Below we run the model for 10 years” but then the variable years is set to 50 (with a comment saying run for 10 years).

Reviewer #2: Review of PONE-D-19-20184- “LeMaRns: a Length-based Multispecies analysis by numerical simulation in R

This manuscript introduces an R version of the ecosystem model LeMans first developed by Hall et al. 2006. The authors provide a very brief introduction to the model and a few examples of ways the model can be applied. The manuscript is well written but falls very short of any actual detail. The authors have decided to push almost all of the meat of this study into the supplemental materials. Although this may be what the journal wants, as a potential end-user of the package I find it very frustrating to not have the most pertinent information in the main text. As a result, the manuscript reads more like a vignette that could be included with the package on CRAN rather than the primary literature source for the model.

The statement on lines 108 – 110 concerns me quite a bit. It sounds like it is saying that the R package can not reproduce the LeMans outputs using the same data set? If so, that is not good. If a few more parameters need to be adjusted so that it will produce the same results than they should be included. The premise of the R package is that it is a easier more transparent version of LeMans but that may not be the case it it can't reproduce the results using the same data.

Another small editorial note, I wonder why the title of the package is LeMaRns, which stands for a Length-based Multi-species analysis by numerical simulation in R. Why not LeMansR? Not only is LeMansR easier to say it makes it obvious that it is an R version of LeMans and not a completely different model.

Other detailed notes:

Line 40 – 41 – claim that LeMans is less complicated than Ecopath or Atlantis. It is true that it is less complicated than Atlantis but Ecopath does not require length data, stock-recruitment relationships, or catchability information.

Line 66 – 69 – I don't think this is necessary to describe the model unless stepping through all of the code. A simple mention tht it is available of CRAN in the introduction should be sufficient.

Line 75 – Are you using the specialized or general k form of the von Bertalanffy growth equation (See Essington et al. 2001 CJFAS 58(11): 2129-2138)?

Line 85 – States that users may input a predator-prey matrix. Is there a default if they do not?

Line 96 – Effort and catchability need to be applied to a biomass or number which I assume is the Lj but the way it is written is not clear. Also, should the catchability term be specific to the length group so qkij instead of qik?

Line 102 – A better sub-heading would be “Test Data Set”

Line 103 – While I'm sure the test data is in a data frame that is probably too technical and should just be referred to as the data set instead. It would also be helpful to flesh out why the test data set is included.

Line 121 – What are the default values based on?

Lines 122 – 125 – Which is the default and why?

Line 139 – I'm guilty of this as well but it is bade form to use the same name for a variable as an argument (e.g. effort = effort).

Line 177 – 178 – Is this an expected result based on other Nash equilibrium studies?

Line 213 – I'm not sure that this manuscript is as transparent as the authors intended. Other than calculations for Nash equilibrium there are no equations.

6. PLOS authors have the option to publish the peer review history of their article (what does this mean?). If published, this will include your full peer review and any attached files.

Reviewer #1: Yes: Finlay Scott

Reviewer #2: Yes: Sean M. Lucey

---

## [Decision Letter · Decision Letter 1]

26 Nov 2019

PONE-D-19-20184R1

LeMaRns: a Length-based Multi-species analysis by numerical simulation in R

PLOS ONE

Dear Mr Spence,

Thank you for submitting your manuscript to PLOS ONE. After careful consideration, we feel that it has merit but does not fully meet PLOS ONE’s publication criteria as it currently stands. Therefore, we invite you to submit a revised version of the manuscript that addresses the few minor points raised during the review process.

We would appreciate receiving your revised manuscript by Jan 10 2020 11:59PM. To enhance the reproducibility of your results, we recommend that if applicable you deposit your laboratory protocols in protocols.io, where a protocol can be assigned its own identifier (DOI) such that it can be cited independently in the future. For instructions see: http://journals.plos.org/plosone/s/submission-guidelines#loc-laboratory-protocols

We look forward to receiving your revised manuscript.

Kind regards,

Athanassios C. Tsikliras

Academic Editor

PLOS ONE

Reviewers' comments:

Reviewer's Responses to Questions

**Comments to the Author**

1. If the authors have adequately addressed your comments raised in a previous round of review and you feel that this manuscript is now acceptable for publication, you may indicate that here to bypass the “Comments to the Author” section, enter your conflict of interest statement in the “Confidential to Editor” section, and submit your "Accept" recommendation.

Reviewer #1: All comments have been addressed

Reviewer #2: All comments have been addressed

2. Is the manuscript technically sound, and do the data support the conclusions?

Reviewer #1: Yes

Reviewer #2: Yes

3. Has the statistical analysis been performed appropriately and rigorously? 

Reviewer #1: Yes

Reviewer #2: Yes

4. Have the authors made all data underlying the findings in their manuscript fully available?

Reviewer #1: Yes

Reviewer #2: Yes

5. Is the manuscript presented in an intelligible fashion and written in standard English?

Reviewer #1: Yes

Reviewer #2: Yes

6. Review Comments to the Author

Reviewer #1: (No Response)

Reviewer #2: Review of PONE-D-19-20184-R1 “LeMaRns: a Length-based Multispecies analysis by numerical simulation in R

This is a revision of the manuscript that introduces an R version of the ecosystem model LeMans. The authors have done a good job addressing my major concerns from the first review. My biggest concern was the inability of the package to recreate a published version of the LeMans model but that has been resolved. I would still prefer to see more meat of the package in the main text rather than as supplemental but see the type of article the authors are going for.

I do have a few minor editorial comments:

Abstract – final sentence says “...leading to increase uptake by fisheries management community”. This is an optimistic statement that should say “we hope will lead to...”. Which is how the authors say it in the main text (Line 36)

Line 63 – Can you provide a citation for how spatial information is included implicitly?

Line 78 – States that predation mortality is size- and species-dependent. I think there is a term missing from M2. Right now there is only a i (species), j (length), and t (time). Nothing for predator or predator size.

Fishing mortality equation has a number but none of the other equations do.

Line 122 – delete “(the default option is hockey-stick)” as it is already stated earlier in the sentence.

Line 148 – It is unclear how NS_other has been calibrated to represent other food. I understand that this was how the authors dealt with my concern over LeMaRns not recreating a published version but this needs further explaination.

Line 170 – The statement about the model is run for 10 years with no fishing can be confusing. Better to note that the default parameters are 10 years and 0 fishing effort.

Line 179 – What is the units for effort? Is this relative effort?

Conclusion – The authors should add a paragraph echoing the end of the abstract/ intro to tie everything together.

7. PLOS authors have the option to publish the peer review history of their article (what does this mean?). If published, this will include your full peer review and any attached files.

Reviewer #1: Yes: Finlay Scott

Reviewer #2: Yes: Sean M. Lucey

---

## [Editor Report · Decision Letter 2]

30 Dec 2019

LeMaRns: a Length-based Multi-species analysis by numerical simulation in R

PONE-D-19-20184R2

Dear Dr. Spence,

We are pleased to inform you that your manuscript has been judged scientifically suitable for publication and will be formally accepted for publication once it complies with all outstanding technical requirements.

With kind regards,

Athanassios C. Tsikliras

Academic Editor

PLOS ONE
---

## [Editor Report · Acceptance letter]

8 Jan 2020

PONE-D-19-20184R2 

LeMaRns: a Length-based Multi-species analysis by numerical simulation in R 

Dear Dr. Spence:

I am pleased to inform you that your manuscript has been deemed suitable for publication in PLOS ONE. Congratulations! Your manuscript is now with our production department. 

With kind regards,

on behalf of

Dr. Athanassios C. Tsikliras 

Academic Editor

PLOS ONE